

# Comparing hand-based and controller-based interactions in virtual reality learning: effects on presence and interaction performance

Murat Saran

Computer Engineering, Cankaya University, Ankara, Turkey

## ABSTRACT

Virtual reality (VR) holds significant promise for enhancing science education by providing immersive and interactive learning experiences. However, the optimal interaction modality within educational VR environments remains an open question. This study investigates the impact of hand-based *vs.* controller-based interaction on sixth-grade students' sense of presence and interaction performance in a VR science laboratory simulation. Fifty-four sixth-grade students were randomly assigned to either a hand-based interaction group or a controller-based interaction group. Participants completed three interactive science experiments (solar system, electrical circuits, and force/energy) within a virtual laboratory environment designed to mimic their school's physical lab. Presence was assessed using a validated Turkish adaptation of the Presence Questionnaire (PQ), while interaction performance was evaluated using a structured observation form completed by a school teacher. Independent samples t-tests and Mann-Whitney U tests were used to compare the presence and performance scores between the groups. Supplementary analyses explored the effects of gender and prior VR experience. Contrary to expectations, no significant differences were found in either presence ($t(49.4) = -0.01$, $p = 0.992$) or interaction performance ($t(52) = -1.30$, $p = 0.199$) between the hand-based and controller-based interaction groups. Both interaction modalities yielded comparable levels of self-reported presence and observed performance. However, an unexpected finding emerged regarding performance. A supplementary analysis revealed a significant main effect of gender on performance scores ($F(1, 50) = 4.844$, $p = 0.032$), independent of interaction type. Specifically, males demonstrated significantly higher performance than females. This study suggests that, for sixth-grade students engaging in these specific VR science simulations, hand-based and controller-based interactions are equally effective in terms of fostering presence and supporting interaction performance. These findings have practical implications for the design and implementation of VR learning environments, particularly in resource-constrained settings where the reduced maintenance and hygiene concerns associated with hand-based interaction may be advantageous.

Corresponding author
Murat Saran, saran@cankaya.edu.tr

## INTRODUCTION

Virtual reality (VR) is a digitally created environment that provides advanced, immersive, and engaging interactions in a three-dimensional space. According to *Sherman & Craig (2019)*, VR is defined as follows:

> "a medium composed of interactive computer simulations that sense the participant's position and actions and replace or augment the feedback in one or more senses, giving the feeling of being mentally immersed or present in the simulation (a virtual world)" (p. 16).

This definition highlights the core principle of VR: creating a sense of presence and making users feel as if they are truly inside the virtual environment. *Sherman & Craig (2019)* further elaborates on this by highlighting five key elements of the VR experience. The first element is the **participants**, as VR is fundamentally about the experience of those engaged with the medium. Each participant brings their background and interpretation, shaping their unique virtual world experience. The second element is the **creators**, who design and develop VR content, ensuring that it provides an engaging and immersive experience. The third element is the **virtual world**, which refers to the simulated environment with which the participants interact. It can exist independently of any display system and is the foundation for VR experiences. The fourth element is **immersion**, which is crucial to creating a compelling VR experience. Immersion can be mental, where users feel emotionally involved, and physical, where sensory input is replaced or augmented by technology. Lastly, **interactivity** is essential, as it allows users to influence their experience within the virtual world. Unlike traditional media, VR responds dynamically to participant actions, enabling real-time interaction and a sense of agency. These elements work together to create a compelling and immersive VR experience.

Since the launch of major commercial products in 2016, the adoption of VR has grown significantly (*Sherman & Craig, 2019*). This expansion is fueled by rapid hardware advancements, including wireless headsets, higher resolution displays, lower latency, and integrated hand tracking, which have made technology more powerful and accessible (*LaValle, 2023*). Consequently, VR is moving beyond its roots in entertainment and finding practical applications in diverse sectors such as healthcare, industrial design, and education (*Radianti et al., 2020*). This trend is mirrored in academia, where research interest has surged; for instance, the annual number of publications related to VR in the Core Collection of the Web of Science has increased from 824 in 2015 to 5,498 in 2024 (Fig. 1).

In education specifically, VR offers the potential to create interactive and lifelike experiences that enhance student engagement by allowing them to explore and manipulate simulated environments (*Slater, 2018*; *Stojšić, Ivkov-Džigurski & Maričić, 2019*; *Chen et al., 2024*; *Lin et al., 2024*). A critical factor in the effectiveness of VR-based learning is the user's sense of presence, the feeling of being immersed in the virtual environment, which is significantly influenced by the quality of interaction within the virtual world (*Slater, 2018*).

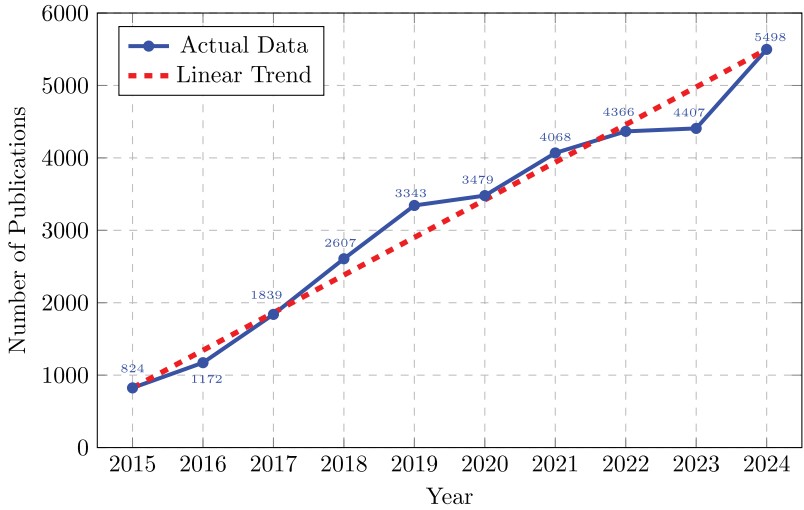

**Figure 1 Number of academic publications on VR technologies from 2015 to 2024.** Data retrieved from Web of Science Core Collection using title-based search for "virtual reality" or "VR" keywords.

This study focuses on investigating how hand-based and controller-based interactions impact the user's sense of presence in VR-based learning environments.

While presence is theoretically important for VR effectiveness, empirical research on VR's educational impact reveals a more nuanced picture. The overall impact of VR on educational outcomes remains an active area of research with mixed findings. Although meta-analyses have shown positive effects on learning outcomes and engagement, particularly in K-6[1] education (*Villena-Taranilla et al., 2022*) and at various educational levels (*Yu & Xu, 2022*; *Yang et al., 2024*), some studies indicate that VR can increase cognitive load, potentially affecting learning performance (*Parong & Mayer, 2021*). Given this complexity in research findings, understanding how different interaction modalities affect presence becomes particularly relevant, as the influence of VR on student learning varies depending on the context and design of the VR experience.

In this study, sixth-grade students used a VR simulation with interactive science experiments on the solar system, electrical circuits, and force and energy. These experiments were designed in collaboration with middle school science teachers on the basis of the expected learning outcomes of the students. In the study, participants were randomly assigned to one of two groups: one group used handheld controllers, while the other group used hand-tracking technology to interact with the virtual learning environment. Depending on the group they were assigned to, participants interacted with objects in the VR environment by pressing the trigger button of the VR controller or moving their hands in the air and making a pinch gesture in the hand interaction method. The study examines how these different interaction methods affect students' sense of presence and interaction performance in the virtual learning environment. The findings of this research will provide valuable information for developers, helping them design more

[1] K-6 refers to the period of schooling from kindergarten through 6th grade.

effective and engaging VR learning experiences that maximize user engagement and learning potential.

The effectiveness of VR-based learning depends on several factors, including content quality, the design of the virtual environment, the user's sense of presence, and the methods of interaction within the virtual environment. To explore these dimensions, the theoretical foundation of this study is based on three main areas, progressing from general to specific. First, the impacts of VR on student learning are examined, presenting its advantages and challenges within the educational context. Next, studies on presence, a critical factor that influences user engagement and learning outcomes in VR, are summarized. Finally, studies comparing hand- and controller-based interactions in VR are reviewed and how these interaction methods affect user experience and interaction performance is analyzed.

## Designing for user experience in educational VR

VR in education has attracted significant interest due to its potential to provide immersive and interactive experiences that are unattainable with traditional methods (*Bailenson, 2018*; *Parong & Mayer, 2018*; *Kamel Boulos, Hetherington & Wheeler, 2008*; *Dalgarno & Lee, 2010*). However, creating an effective user experience that facilitates engagement and seamless interaction is a complex design challenge. The design of the VR environment, including its instructional elements and interaction mechanics, directly influences cognitive load, user perception, and engagement. This section reviews key findings that highlight the factors shaping the user experience in educational VR, providing a foundation for understanding the critical role of interaction modalities.

The immersive characteristics of VR exert a bidirectional influence on user cognitive processes. On the one hand, VR can improve engagement and specific cognitive skills, particularly in immersive settings and for certain disciplines such as medicine (*Yu & Xu, 2022*; *Yang et al., 2024*). However, it can increase cognitive load, lead to disorientation, and may not always result in better learning outcomes compared to traditional methods, especially if not properly designed. For example, *Parong & Mayer (2021)* compared a VR biology session with a PowerPoint presentation and found that while VR increased enjoyment, it also led to a higher extraneous cognitive load. This suggests that the rich, immersive details of VR, while boosting presence, can act as 'seductive details' that distract users and interfere with task performance (*Parong & Mayer, 2021*). This finding underscores the importance of careful instructional design that minimizes cognitive overload to ensure that the user can focus on the intended interaction and content (*Mayer, 2001*). Similarly, the research by *Fitton, Finnegan & Proulx (2020)* found that immersive virtual environments could match the knowledge retention of traditional classrooms, producing similar levels of student satisfaction and engagement. This indicates that the mere use of VR is not a panacea; its effectiveness hinges on thoughtful design choices.

Beyond cognitive factors, the design of interactive elements within the VR environment is crucial to user engagement and acceptance. Although educators perceive VR as a motivating tool that can illustrate abstract concepts and transcend geographical barriers (*Orhak & Çağıltay, 2024*; *Stojšić, Ivkov-Džigurski & Maričić, 2019*), the details of the

implementation determine its success. For example, the design of embodied agents (EA) can significantly affect the user experience. *Fitton, Finnegan & Proulx (2020)* noted that 58% of the participants in the condition with the humanoid tutor found the avatar to be "strange," citing its repetitive movement and lack of facial expressions. The authors explicitly suggest that, unlike the abstract tutor, the humanoid avatar was susceptible to the uncanny valley effect. This highlights that every design element, from avatars to interface mechanics, contributes to the overall user experience and can either enhance or detract from engagement.

Practical and usability challenges must also be addressed to ensure a positive user experience. Reviews and meta-analyses point to persistent barriers, such as high costs and technical complexity (*Jensen & Konradsen, 2018*; *Al-Ansi et al., 2023*; *Begum et al., 2024*). The review by *Jensen & Konradsen (2018)* also highlights the "widespread cybersickness" as a significant issue. These challenges can undermine the potential benefits of VR technology by disrupting or distracting from the learning task. Overcoming them requires not only technical solutions, but also integrating user-centered design principles to create intuitive and accessible experiences (*Mallek et al., 2024*).

In summary, the literature indicates that the user experience in educational VR is complex. Although VR offers powerful opportunities for immersion and engagement, its effectiveness is contingent on a design that manages cognitive load, provides intuitive interactive elements, and mitigates usability issues. A fundamental component of this design is the method of interaction. This study, therefore, narrows its focus to compare hand- and controller-based interactions, seeking to understand how this critical design choice impacts presence and interaction performance within a VR environment.

## The construct of presence

The concept of "presence" in virtual environments (VEs) - defined as the subjective experience of "being there" *Witmer & Singer (1998)* - has been the cornerstone of VR research. The foundational work of Witmer and Singer conceptualized the presence as emerging from the interplay of user involvement and technological immersion, which they described as the "perceptual envelope" of the VE. Their development of the Presence Questionnaire (PQ) provided a methodological tool to quantify this experience, linking higher presence scores to improved task performance and reduced simulator sickness, and identifying factors such as system control, sensory fidelity, and realism as key determinants (*Witmer & Singer, 1998*).

To refine this framework, contemporary research often distinguishes presence from immersion. *Slater (2018)* proposes a highly influential distinction where immersion is an objective, technological property of a system (*e.g.*, display resolution, field of view, tracking accuracy), whereas presence is the resulting subjective, psychological sensation of being in the virtual place. From this perspective, presence is not a belief that the virtual world is real, but rather a perceptual illusion maintained by plausible sensorimotor contingencies: the system responding to a user's actions in a way that is expected and coherent.

This focus on sensorimotor contingencies brings the user's virtual body, or avatar, to the forefront. When users are represented by an avatar, their experience is not just about being

in a place, but about being in a body within that place. This is captured by the sense of embodiment (SoE), a related but distinct construct from presence. *Kilteni, Groten & Slater (2012)* deconstruct SoE into three critical subcomponents:

**The sense of self-location:** the feeling of being located inside the physical space of the virtual body. This is strongly influenced by a first-person visual perspective.

**The sense of agency:** the feeling of having motor control over the virtual body, where its movements are a direct consequence of one's own intentions and actions.

**The sense of body ownership:** the feeling that the virtual body is one's own, a process of self-attribution that is strengthened by synchronous multisensory feedback (*e.g.*, visuomotor or visuotactile correlations) and the morphological similarity of the avatar to a human form.

A strong sense of agency is fundamental to effective and intuitive interaction, while the resulting sense of ownership contributes significantly to the general believability of the experience. These components of embodiment are powerful drivers of presence; feeling authentically located within and in control of a virtual body greatly enhances the higher-level illusion of "being there" (*Kilteni, Groten & Slater, 2012*; *Slater, 2009*).

The interplay between presence, embodiment, and user outcomes is well documented. For example, enhanced visual immersion *via* VR headsets improves both presence and performance in training simulators (*Stevens & Kincaid, 2015*), likely because it strengthens the sense of self-location and agency. In contrast, poor interface quality can disrupt agency and ownership, leading to a break in presence and an increase in simulator sickness (*Nichols, Haldane & Wilson, 2000*). The psychological impact extends to the emotional and social domains. Fear and anxiety feelings in virtual environments are related to perceptual plausibility and the user's embodied state (*Diemer et al., 2015*). Furthermore, social presence, the feeling of being with others, is significantly enhanced by embodied avatars, even minimal ones, and can increase a user's compliance with the suggestions of a virtual agent, highlighting the persuasive power of embodied interaction (*Erickson-Davis et al., 2021*).

Integrating these perspectives, it becomes clear that presence is a multifaceted construct. It rests on a technological foundation of immersion, but emerges psychologically from the interplay of place illusion (the sense of being in the VE) and the sense of embodiment (the sense of being, owning, and controlling a body within that VE). When other agents are present, the social presence adds another critical layer. For practical applications, VR design must therefore balance sensory realism with interactive fidelity, user agency, and psychological engagement to foster meaningful and effective user-environment interactions.

## Hand-based *vs.* controller-based interactions in VR

The main challenge facing VR systems is the creation of interaction methods that effectively merge functionality and naturalness. Although traditional methods often utilize handheld controllers, providing consistent input and haptic feedback for improved

accuracy and immersion (*Dias et al., 2017*; *Caggianese, Gallo & Neroni, 2019*), they do so by employing an intermediary rather than allowing direct interaction with the virtual environment (VE). Advances in sensor technology and computer vision have made it possible to use hand-based interaction, allowing natural gestures within VR (*Caggianese, Gallo & Neroni, 2019*; *Khundam et al., 2021*). This approach improves intuitiveness and the sense of embodiment by aligning physical actions with virtual experiences through embodied cognition (*Lin et al., 2019*; *Lou et al., 2021*; *Pangestu et al., 2021*; *Johnson-Glenberg, 2018*). However, hand-based interactions face obstacles such as unstable tracking, decreased precision, and lack of tactile feedback (*Argelaguet et al., 2016*; *Kilteni, Groten & Slater, 2012*; *Tong et al., 2023*).

Empirical studies of these modalities yield varied results, with controllers often outperforming in precision-dependent tasks. Research indicates that participants achieved faster and more precise results in object manipulation and medical simulations using handheld devices compared to hand tracking (*Caggianese, Gallo & Neroni, 2019*; *Khundam et al., 2021*). In VR medical training, specifically intubation, interaction time and usability did not show significant differences between controllers and hand tracking, but controllers were slightly favored for ease (*Khundam et al., 2021*). In reach-pick-place studies, controllers were associated with reduced play durations and trigger frequencies (*Hameed, Perkis & Möller, 2021*). In contrast, hand-based interactions excel in creating a strong sense of ownership and user preference, especially in touch-oriented tasks (*Lou et al., 2021*; *Luong et al., 2023*). Free-hand input is preferred for its intuitive nature in direct tasks, while controllers are preferred for ray casting due to better precision and less effort (*Luong et al., 2023*). This highlights a task-dependent efficacy: controllers excel in precision tasks, while hand tracking is better for enhancing agency and realism.

Interaction methods further complicate the dichotomy in user experience. Hand-based interaction, praised for its natural feel due to alignment with the real-world motor schema, enhances immersion (*Lin et al., 2019*; *Argelaguet et al., 2016*; *Almeida et al., 2019*). Gesture-based VR systems boost engagement and educational outcomes by engaging sensorimotor pathways (*Johnson-Glenberg, 2018*). For example, a cyber glove system increased the sense of embodiment and naturalness in interaction tasks, with 90% of users noting an equal or increased embodiment (*Almeida et al., 2019*). However, this preference is not universal; controllers are preferred for ray-casting tasks due to greater familiarity and precision (*Lou et al., 2021*; *Luong et al., 2023*). Moreover, the absence of standardized gestures and cumbersome devices presents challenges for gesture interaction (*Li et al., 2019*). This variability accentuates how the benefits of the task and previous user experiences, such as familiarity with gaming controllers, influence perceived ease of use (*Khundam et al., 2021*).

Technological limitations are cited in the literature as a critical barrier to the widespread adoption of hand tracking. Issues such as latency, limited field of view, and intermittent tracking errors can diminish user agency and satisfaction (*Khundam et al., 2021*; *Pangestu et al., 2021*; *Tong et al., 2023*). These limitations accentuate the need for advances in sensor accuracy and algorithmic robustness. Furthermore, integrating haptic feedback into hand-based systems presents a key research frontier, as its absence currently undermines

the tactile realism essential for fully immersive interactions (*Argelaguet et al., 2016*; *Tong et al., 2023*). Current studies support hand-based interaction with rigid objects well, but interactions with complex material objects, such as clay-like objects and fluids, suffer from low update rates due to computational complexity (*Tong et al., 2023*). Addressing these challenges is crucial for developing robust hand-based haptic interactive platforms (*Tong et al., 2023*).

In essence, selecting between hand and controller interactions in VR depends on the task needs and user experience goals. Controllers deliver precision and are user-friendly, while hand interactions offer a natural feel that is advantageous for education and immersion. Recognizing the importance of natural interaction in educational environments, this study investigates how hand-tracking interaction in an educational virtual science lab simulation affects students' experience, aiming to increase presence and interaction performance in a VR environment.

## Purpose and questions of the research

This research investigates how using hands *vs.* controllers influences presence and interaction performance in an educational VR simulation. The simulation includes interactive science experiments on the solar system, electrical circuits, and force and energy, offering a platform to explore immersive and interactive learning. The increasing use of VR in education is due to its ability to provide experiences that traditional methods cannot replicate: involving students in environments that may be hazardous, expensive, or rare. However, the success of VR in educational settings depends on critical factors such as **presence**—the subjective sensation of "being there" in the virtual environment—and **interaction performance**, which measures the efficiency and precision of user actions within the simulation. These factors are shaped by the interaction modality employed, necessitating a comparative analysis of hand-based and controller-based approaches.

Building on the literature exploring the design principles for an effective user experience in educational VR (*Mayer, 2001*; *Fitton, Finnegan & Proulx, 2020*; *Mallek et al., 2024*), the concept of presence, and the effectiveness of different interaction modalities, this study proposes two hypotheses to investigate how these methods influence presence and performance. These hypotheses are informed by empirical evidence and theoretical frameworks that emphasize the critical role of interaction design in shaping cognitive load (*Parong & Mayer, 2021*), user engagement (*Orhak & Çağıltay, 2024*; *Stojšić, Ivkov-Džigurski & Maričić, 2019*), and overall VR effectiveness (*Yu & Xu, 2022*; *Jensen & Konradsen, 2018*), establishing a solid foundation for the anticipated results.

### Research questions (RQs)

The study aims to answer the following questions.

1. How do hand-based and controller-based interactions influence the sense of presence in a VR learning environment?
2. How do these interaction modalities affect interaction performance in educational VR simulations?

The findings seek to deepen the understanding of optimal interaction design for VR-based educational tools, striking a balance between the immersive advantages of presence and the practical requirements of effective performance. This research builds on the literature's acknowledgment of VR's potential to enrich learning through engagement and immersion, while also tackling challenges such as cognitive load, precision, and technical constraints identified in prior studies.

### Hypotheses

Based on the literature review and research objectives, to answer the RQs, the following hypotheses were formulated:

1. **Hypothesis 1 (H1):** hand-based interaction will result in significantly higher presence scores than controller-based interaction.

Hand-based interactions mirror natural motor schemas and activate sensorimotor pathways, creating more intuitive engagement with virtual environments (*Argelaguet et al., 2016*; *Johnson-Glenberg, 2018*; *Lin et al., 2019*; *Scavarelli, Arya & Teather, 2021*). This enhanced alignment between physical movements and virtual experiences is expected to strengthen the subjective sense of "being there" in the VR environment, potentially leading to deeper engagement with educational content.

2. **Hypothesis 2 (H2):** the controller-based interaction will result in significantly better interaction performance than the hand-based interaction.

Controllers provide consistent input precision and enable faster and more accurate task completion in VR environments (*Caggianese, Gallo & Neroni, 2019*; *Khundam et al., 2021*; *Hameed, Perkis & Möller, 2021*; *Luong et al., 2023*). This technical advantage is particularly beneficial for structured educational tasks that require precise object manipulation, efficient interaction sequences, and reliable input recognition. The standardized button mappings and haptic feedback from controllers facilitate reduced task completion times and fewer interaction errors, which are critical to maintaining learning flow and preventing frustration in educational contexts.

This study involved sixth grade students participating in interactive science experiments on the solar system, electrical circuits, and force and energy within a VR simulation designed in collaboration with middle school science teachers. Participants were randomly assigned to a hand- or controller-based interaction group. Contrary to expectations, the results revealed no significant differences in the presence or performance of the interaction between the two interaction modalities. However, an unexpected finding showed a significant gender effect, with males demonstrating higher performance scores than females, regardless of the type of interaction. The following sections outline the study methodology, present the findings, and discuss their implications for designing effective VR learning environments.

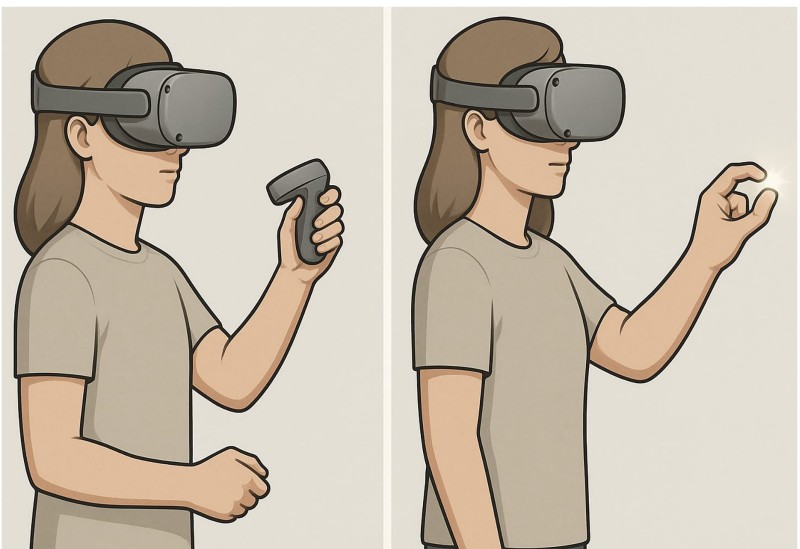

**Figure 2** **Controller _vs._ hand-based interactions.** This original illustration was created specifically for this research.     

# METHODS

## Research design and procedure

A between-subjects experimental design was used to investigate whether the type of interaction affects user presence and performance. Fifty-four sixth-grade students from a private secondary school participated in the study. The students were randomly divided into two groups, corresponding to the two levels of the independent variable, Interaction Type: (1) VR controllers and (2) direct interaction _via_ VR hand-tracking. Through random assignment, balanced groups were ensured, including gender balance, and selection bias was minimized. The study ensured gender balance among participants to promote equitable access to these learning experiences. The hand tracking group interacted with hand gestures, while the controllers group used a physical controller (Fig. 2). Task performance and self-reported presence were measured as dependent variables. In all other aspects, the VR environment and content were kept constant in both conditions.

The experimental procedure was identical for both groups. Before the session, all participants received a brief introduction to the interaction commands for their assigned condition (hand gestures or controllers). Each participant was assigned a total of 30 min to perform three different experiments within the VR lab application, with a maximum of 10 min allowed for each experiment. This 30-min duration refers strictly to VR tasks, excluding the brief introduction to interaction commands and the transition to the classroom for questionnaire completion. This research used Meta Oculus Quest 3, a standalone VR headset with dual LCD screens each offering a resolution of $2,064 \times 2,208$ pixels, a 110 degree horizontal and 96 degree vertical field of view, and a maximum refresh rate of 120 Hz. The device is powered by the Qualcomm Snapdragon XR2 Gen 2 processor with 8GB of RAM, enabling high-performance graphics and smooth interactions.

It supports 6-degree-of-freedom (DOF) tracking through four integrated infrared (IR) tracking cameras and a depth sensor, facilitating precise positional tracking of the headset and Touch Plus controllers. Additionally, Meta Quest 3 includes full-color pass-through cameras for mixed reality experiences, enhancing environmental awareness, and advanced hand-tracking capabilities for controller-free interactions (*Meta, 2023*).

To assess subjective experience, each participant was guided to an empty classroom immediately after the session to complete a paper-based VR presence questionnaire. This procedure ensured a quiet environment free from distractions. Performance was evaluated through structured observation, as task completion time was not a recorded metric in this study. A school teacher monitored each participant using a pre-defined observation form in conjunction with the Meta Horizon mobile application, which provided a real-time view of the student's perspective. This method focused on assessing the quality of the student interactions and their success in completing the tasks. The teacher's observation form and the VR presence questionnaire are included in the Supplemental Materials. Figure 3 illustrates the comprehensive experimental design.

### Participants

Fifty-four sixth-grade students (ages 11–12) from a private secondary school in Türkiye participated in this study. Participants were randomly assigned to one of two interaction conditions while maintaining gender balance: hand-based interaction ($n = 27$; 13 males, 14 females) and controller-based interaction ($n = 27$; 14 males, 13 females). The stratified random assignment ensured an approximately equal gender distribution between the conditions, as shown in Table 1.

The sample size was determined based on practical constraints on participant availability at the school and the statistical power required to detect differences between two independent groups using the chosen statistical tests. All participants were novices to the specific VR lab application used in the study, although some had prior general VR experience, which was later analyzed as a supplementary variable. Participation required a signed parent consent form, which included brief information about the study and was distributed to parents one week before the experiment. In addition, approval for the study was obtained from the Ethics Committee of Cankaya University (Application Ref: 31.05.2025/9), ensuring compliance with ethical standards for research involving children.

### VR-based laboratory simulations

In this study, virtual laboratory simulations were developed using the Unity VR framework (*Unity, 2025*) to improve science learning among middle school students. Preliminary interviews with teachers revealed that students faced difficulties understanding key concepts related to the solar system, electrical circuits, and force and energy. Consequently, these topics were selected as the focus for the simulations.

The instructional design of this study is explicitly grounded in Kolb's Experiential Learning Theory (ELT), a well-established theoretical framework that conceptualizes learning as a cyclical process of experience, reflection, conceptualization and experimentation (*Kolb & Kolb, 2017*). ELT provides a robust foundation for

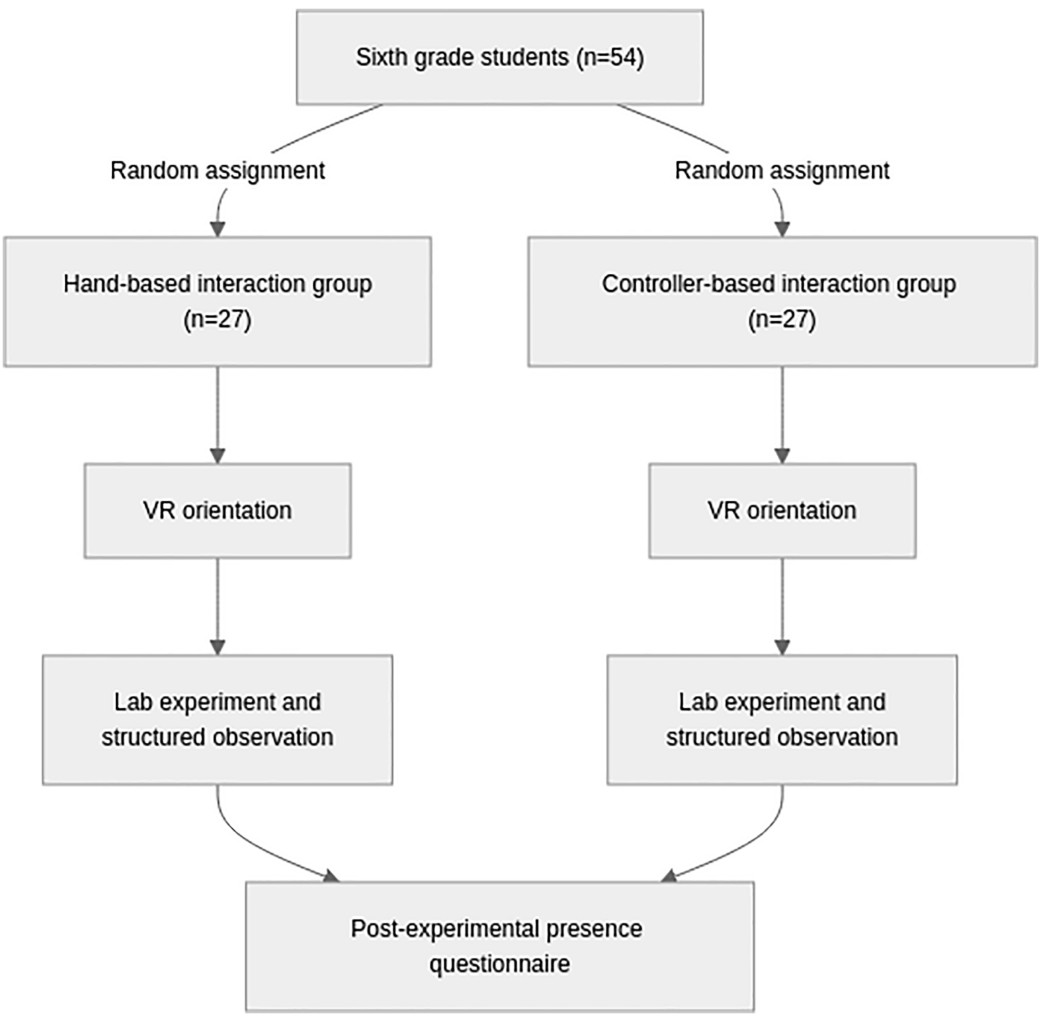

**Figure 3 Overall experimental design.**

**Table 1 Participant distribution in interaction groups.**

| Group | Males (n) | Females (n) | Total (n) |
|---|---|---|---|
| Hand-based interaction | 13 | 14 | 27 |
| Controller-based interaction | 14 | 13 | 27 |

simulation-based learning environments, as it emphasizes the transformation of experience into knowledge through four interconnected stages: (1) *Concrete Experience*–direct engagement with activities or phenomena; (2) *Reflective Observation*–critical examination of experiences and outcomes; (3) *Abstract Conceptualization*–formation of theoretical understanding and generalizations; and (4) *Active Experimentation*–application and testing of newly acquired knowledge in novel contexts (*Kolb & Kolb, 2017*). This cyclical model recognizes learning as a dynamic and non-linear process that integrates perception, cognition, and behavioral adaptation, making it

particularly suitable for virtual simulation environments where students can safely experiment, observe consequences, and refine their understanding iteratively. The learning objectives for each simulation module are systematically aligned with ELT's four-stage cycle to ensure coherent pedagogical design.

The virtual environment was designed as a 3D model of the school's laboratory to foster a sense of familiarity and immersion, enabling students to feel like they were conducting experiments in their own laboratory. Laboratory components, including tables, chairs, and boards, were meticulously modeled to closely resemble their real-world counterparts. This attention to detail was intended to strengthen the students' sense of presence and engagement within the virtual environment.

Furthermore, an expert graphic designer designed virtual experiment assets, such as scientific instruments and interactive elements, to ensure accuracy and visual fidelity. The high-fidelity graphics ensure that scientific phenomena are represented accurately, allowing students to transfer their virtual experiences to real-world understanding. Figures 4A, 4B provide multiple perspectives of the virtual laboratory environment, highlighting its similarity to the physical laboratory.

In these simulations, students use their randomly assigned interaction type—VR controllers or hand-based control - to engage with the virtual environment. The simulations were designed to seamlessly accommodate both interaction methods, ensuring a consistent learning experience regardless of the input type. The following are descriptions of the three simulations.

### Solar system simulation

Students navigate through a 3D scaled model of the solar system, starting from Earth, and can fly to each planet to explore their relative sizes, distances, and orbital paths. The learning objectives for the solar system simulation module are systematically aligned with ELT's four-stage cycle, as listed below:

**Concrete experience:** students will actively explore a three-dimensional solar system model through virtual navigation, directly manipulating viewpoints, and interacting with planetary objects to gain first-hand experience with spatial relationships and celestial mechanics.

**Reflective observation:** students will critically analyze their navigation strategies and system-generated feedback to identify misconceptions regarding planetary sequences, relative distances, and orbital characteristics.

**Abstract conceptualization:** students will synthesize observational data to construct coherent mental models of the organization of the solar system, articulating the principles governing planetary order, relative scales, orbital mechanics, and distinguishing physical characteristics such as atmospheric composition and surface features.

**Active experimentation:** students will validate their conceptual understanding through interactive assessments, including planet sequencing tasks, distance estimation challenges, and predictive exercises in novel navigation scenarios.

(A) View 1                                      (B) View 2

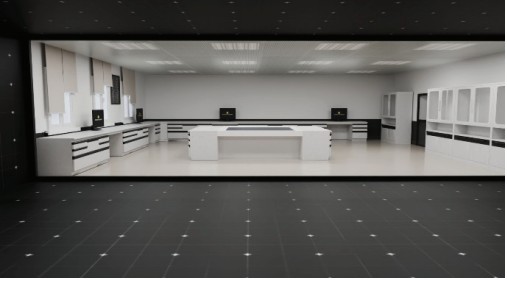
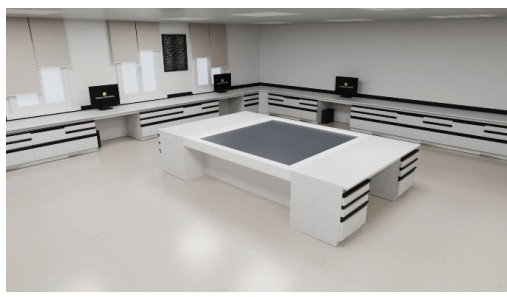

**Figure 4** **Screenshots of virtual laboratory environment.**

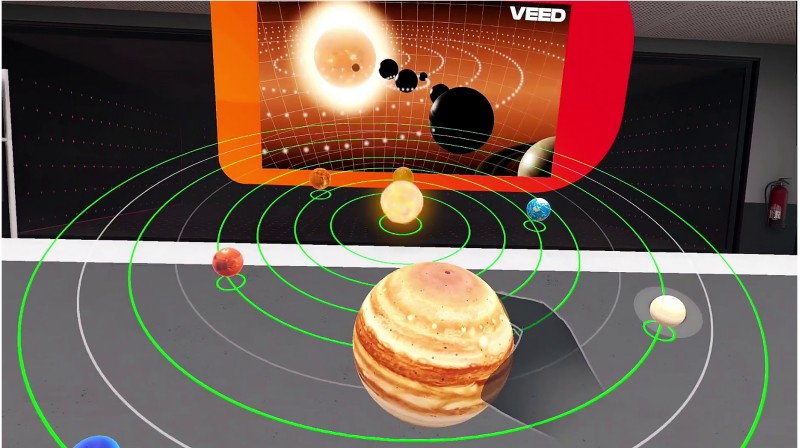

**Figure 5** **Screenshot of the virtual laboratory simulation of the solar system, showcasing the interactive user interface and dynamic visualization of planetary orbits.**

As shown in Fig. 5, the virtual laboratory simulation of the solar system provided participants with an interactive interface to explore the planetary orbits, a key component of the experimental design. They select planets to access interactive labels detailing characteristics such as composition, atmosphere, and notable features (*e.g.*, Jupiter's Great Red Spot). The simulation emphasizes Earth-planet relationships, allowing students to observe accurate spatial arrangements, such as the order of planets from the Sun or their size rankings. If a student navigates incorrectly, gentle corrective feedback guides them, *e.g.*, 'You are heading toward Mars; let us visit Venus first, closer to the Sun.' Additional features include challenges, such as organizing planets by size or distance, and interacting with the atmosphere of the Earth to trigger educational content about its layers and significance, enhancing the retention of key concepts.

### Electric circuit simulation

This simulation allows students to build and test electrical circuits in a virtual environment that mimics a real electronics lab. The learning objectives of the electric

circuit simulation module are systematically aligned with the four stages of the ELT cycle as follows:

**Concrete experience:** students will construct and manipulate virtual electrical circuits using simulated components (batteries, resistors, switches, measurement devices), observing real-time electrical behavior and system responses.

**Reflective observation:** students will analyze discrepancies between predicted and observed circuit behavior, examining system feedback (voltage readings, current flow indicators, component failures) to identify flaws in circuit design or conceptual understanding.

**Abstract conceptualization:** students will develop comprehensive models of electrical current flow, resistance relationships, and energy distribution in series and parallel configurations, formulating predictive principles for circuit behavior under varying conditions.

**Active experimentation:** students will test theoretical predictions through systematic circuit modifications and respond to embedded assessment prompts that require the application of electrical principles to novel circuit configurations.

As shown in Fig. 6, students manipulate virtual circuit components, such as batteries, switches, and light bulbs, to build circuits on a virtual breadboard. They can connect these components in series or parallel configurations and observe the resulting changes in current and voltage. For example, adding a bulb in series dims the lights, while adding one in parallel maintains brightness, providing a clear demonstration of circuit behavior. If a student connects a circuit incorrectly (*e.g.*, leaving it open), the simulation offers visual cues, such as unlit bulbs and a guidance message like 'Check your connections; the circuit is incomplete.' To deepen understanding, mini-quizzes prompt students to predict outcomes, *e.g.*, 'What happens to brightness if you add another bulb in series?', before they test their hypotheses, fostering critical thinking.

### Force and energy simulation

In this simulation, students explore the relationship between force, work, and energy through hands-on experiments. The learning objectives of the force and energy simulation module are systematically aligned with the four phases of the ELT cycle.

**Concrete experience:** students will manipulate virtual mechanical systems (inclined planes, spring mechanisms, pendulums, masses) by applying forces and observing the resultant motion, energy transformations, and equilibrium states.

**Reflective observation:** students will interpret simulation feedback regarding force magnitude, direction, and effectiveness, analyzing unsuccessful attempts to identify gaps between intended actions and physical principles.

**Abstract conceptualization:** students will articulate quantitative and qualitative relationships governing force, work, and energy interactions, including potential kinetic

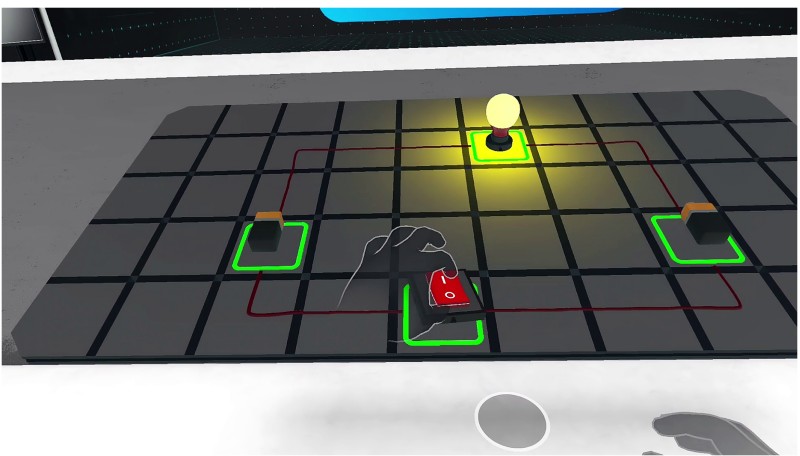

**Figure 6 Screenshot of the electric circuit simulation that illustrates the interactive construction of a circuit with virtual components, including batteries, switches, and light bulbs, on a virtual breadboard that mimics a real electronics lab.**

energy transformations, applications of the work-energy theorem and equilibrium conditions under varying mass and force parameters.

**Active experimentation:** students will apply mechanical principles to solve progressively complex scenarios, predict system behavior, and validate hypotheses through controlled experimentation within the simulation environment.

Students interact with virtual objects, such as pushing a cart up a ramp, to observe how the applied force increases the potential energy of the cart as it gains height. In addition, students can manipulate virtual weights of 20, 30, and 40 kg to investigate force equilibrium and gravitational potential energy, as shown in Fig. 7. Simulation provides real-time feedback, showing metrics such as force applied, distance moved, and work done, and for weights, gravitational force or net force, which helps students visualize these abstract concepts. If a student applies force incorrectly, such as pushing the cart downhill or attempting to lift a weight without sufficient force, the simulation responds with a message like 'This direction does not increase potential energy because the force does not work against gravity' or 'The force applied is insufficient to overcome gravity,' reinforcing a proper understanding. In addition, students can experiment with different objects (*e.g.,* pulling a spring or lifting a weight) to measure energy conversions, the variety of weights enhancing the versatility of the simulation and making it an engaging tool for exploring physics principles in a comprehensive way.

By providing a safe, interactive, and immersive environment, these VR simulations aim to demystify complex scientific concepts, making them accessible and engaging for sixth-grade students. Through hands-on experimentation and immediate feedback, students build a deeper understanding of the solar system, electrical circuits, and force and energy, preparing them for more advanced scientific inquiry.

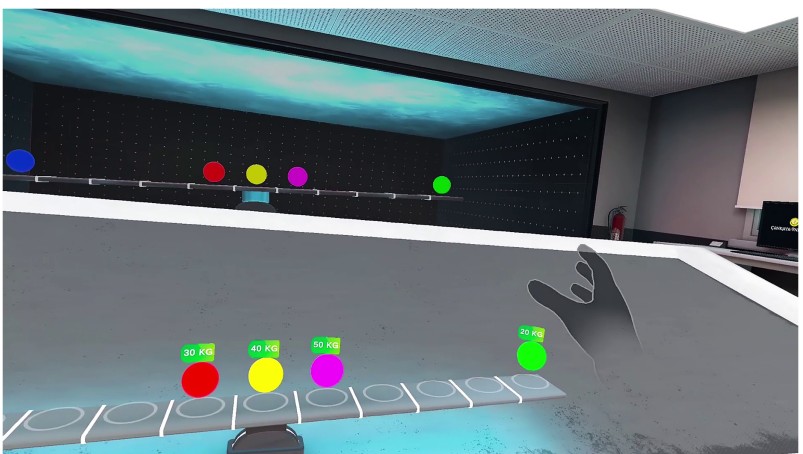

**Figure 7 Screenshot of the virtual physics simulation, illustrating an interactive setup to explore concepts of force and equilibrium.**

## Data collection instruments

### Presence questionnaire

This research utilizes the theoretical model of *Witmer & Singer (1998)*, which conceptualizes presence as a multidimensional subjective experience within a virtual environment. According to this framework, presence emerges from the interaction between technological factors (such as sensory fidelity and interface quality) and user factors (including attention, involvement, and adaptation capabilities).

To assess the sense of presence experienced by participants in the VR learning environment, this study used the Turkish adaptation of the Presence Questionnaire (PQ), originally developed by *Witmer & Singer (1998)*. The PQ is a widely recognized instrument designed to measure the subjective experience of presence in virtual environments, defined as the feeling of being in a virtual space despite physically being elsewhere (*Witmer & Singer, 1998*). The Turkish adaptation, validated by *Gokoglu & Cakiroglu (2019)*, was used to ensure cultural and linguistic suitability for the study's Turkish-speaking participants. This adaptation consists of 29 items and employs a five-point Likert scale (ranging from 1 = "not at all" to 5 = "completely"), adjusted from the original seven-point scale to suit the cognitive level of younger participants in the validation study (*Gokoglu & Cakiroglu, 2019*). The adapted PQ demonstrated strong reliability, with a Cronbach's alpha of 0.844, indicating its suitability as a robust measure of presence (*Gokoglu & Cakiroglu, 2019*).

Given that this study compares two distinct interaction methods, hand-based and controller-based interactions, two versions of the questionnaire were developed to align with these conditions. The 29 core items of the Turkish PQ remained consistent in both versions to ensure comparability in measuring the presence. However, specific questions related to interaction and control were tailored to reflect the input method experienced by participants. For example, items such as "How natural did your interactions with the

environment seem?" were worded in the hand-based version to reference hand movements (*e.g.*, "How natural did your hand interactions with the environment seem?"), while in the controller-based version they referenced controller inputs (*e.g.*, "How natural did your controller interactions with the environment seem?"). This customization ensures that the questionnaire captures the nuances of the presence specific to each interaction method while maintaining the structural integrity of the original scale.

In addition to the standard PQ items, two demographic questions were incorporated into both versions of the questionnaire to gather additional data for analysis. Participants indicated their gender by selecting either "Male" or "Female" using a checkbox format. Participants also responded to the question "Have you ever used VR for games or educational purposes?" with a yes/no checkbox option. These additional questions were included to explore possible variations in presence and interaction performance influenced by gender or prior familiarity with VR.

The questionnaire was administered immediately after each participant completed the VR learning task to capture their immediate impressions, enhancing the precision of their self-reported experiences. Participants completed a paper version of the questionnaire in a quiet area adjacent to the VR setup, ensuring a controlled environment free from distractions. This administration method was chosen for the consistency and ease of implementation within the experimental design of the study.

### Structured observation form

A structured observation form was used to evaluate the performance of the interaction of the participants in the VR learning environment. This form was designed to systematically assess various dimensions of participants' interactions and performance as they engaged with virtual tasks. A single rater, a school teacher familiar with the students and the curriculum, completed the form. The observer completed the form while monitoring the participants' actions in real time through the Meta Horizon mobile application, which provided a live feed of the participant's perspective within the VR environment. This setup enabled for an accurate and detailed evaluation of how participants navigated and interacted with the virtual space.

The structured observation form consists of six dimensions, each focusing on specific aspects of participant experience and performance. Each section employs descriptive rating scales to categorize the participant's proficiency or engagement, providing a clear and consistent framework for evaluation. The following is a list of elements included in the structured observation form:

- **Device handling skill:** rated on a scale from "Very good" (uses the device effortlessly and correctly) to "Poor" (struggles consistently with the device).
- **Following instructions:** assessed on a scale from "Excellent" (follows all instructions accurately and promptly) to "Poor" (fails to follow most instructions).
- **Level of interaction:** evaluated from "Very high" (remains consistently active and engaged) to "Low" (exhibits rare activity or engagement).

- **Participation and interest:** rated from "Very Interested" (displays a strong interest in all activities) to "Uninterested" (shows little or no interest in most activities).
- **Problem-solving skill:** measured on a scale from "Very Good" (resolves problems quickly and effectively) to "Poor" (can not resolve most problems and requires frequent assistance).
- **Overall performance:** holistically assessed with ratings of "Excellent," "Good," "Average." or "Poor."

The form was applied uniformly to both experimental groups, those using hand-based interaction and those using controller-based interaction, ensuring consistency in the assessment process between different interaction methods. By combining quantitative ratings with the use of the real-time observational capabilities of the Meta Horizon mobile application, the structured observation form provided a comprehensive tool to evaluate the performance of participants' interactions in the VR environment.

## Data analysis

Data were analyzed using statistical software R (version 4.4.2). The analysis focused on two primary outcome measures: (1) presence scores, derived from a self-reported presence questionnaire, and (2) interaction performance scores, evaluated using a structured observation form completed by a school teacher. Both instruments were designed to evaluate the key constructs described in the research hypotheses.

**Reliability analysis:** internal consistency of the presence questionnaire and the structured observation form was evaluated using Cronbach's alpha. A threshold of 0.70 or higher was adopted as an indicator of acceptable reliability (*George & Mallery, 2022*).

**Descriptive statistics:** means, standard deviations (SD), and ranges (minimum and maximum values) were calculated for presence and performance scores, reported for each interaction group (based on the hand and the controller) and the overall sample.

**Normality testing:** the Shapiro-Wilk test was applied to assess the normality of the distributions of presence and performance scores within each interaction group. A value of $p$ greater than 0.05 indicated that the data complied with a normal distribution, guiding the selection of parametric or nonparametric tests.

**Group comparison tests:** to test H1 and H2, independent samples $t$-tests were performed to compare mean presence and performance scores between the hand-based and controller-based interaction groups. Levene's tests for equality of variances were performed to ensure that the assumption of homogeneity of variance was met. Effect sizes and 95% confidence intervals were reported to provide additional context for the results.

**Supplementary analyses:** to explore possible modulating variables, independent samples $t$-tests examined the effects of gender and previous VR experience on presence scores. A chi-square test was performed to examine the association between gender and previous VR experience. Furthermore, a two-way analysis of variance (ANOVA) was performed to

investigate interaction effects between interaction type (hand-based *vs.* controller-based) and gender on both presence and performance scores. In addition, covariance analysis (ANCOVA) was performed to examine the effect of gender on performance while controlling for previous VR experience, addressing potential confounding factors.

**Visualization:** boxplots were generated to visually represent the distribution of presence and performance scores between interaction groups. These figures were created in R and exported as PDF files for inclusion in the paper.

## RESULTS

The results are organized into three subsections: (1) the presence questionnaire analysis, (2) the structured observation form analysis, and (3) additional analyses exploring the effects of gender and previous VR experience. All findings are reported with sufficient statistical detail to support interpretation and replication.

### Presence questionnaire analysis

This subsection addresses RQ1: how do hand-based and controller-based interactions influence the sense of presence in a VR learning environment?

**Reliability:** the presence questionnaire exhibited good internal consistency, with a Cronbach's alpha of 0.832, exceeding the acceptable threshold of 0.70.

**Descriptive statistics:** Table 2 summarizes the descriptive statistics for the presence scores by interaction group. The hand interaction group reported a mean presence score of 108.56 ($SD = 14.69$), while the controller interaction group had a nearly identical mean of 108.59 ($SD = 11.65$). The total sample mean was 108.57 ($SD = 13.13$).

**Normality testing:** the Shapiro-Wilk test confirmed that presence scores were normally distributed for both groups: hand-based ($W = 0.963$, $p = 0.424$) and controller-based ($W = 0.959$, $p = 0.355$). Thus, parametric tests were appropriate.

**Group comparison:** to test H1, which predicted that hand-based interaction would produce higher presence scores than controller-based interaction, parametric analyses were performed. An independent samples $t$-test indicated that no significant differences were found in presence scores between the hand-based ($M = 82.15$, $SD = 15.43$) and controller-based ($M = 82.11$, $SD = 12.01$) groups, $t(49.4) = -0.01$, $p = 0.992$, 95% confidence interval (CI) $[-7.29$ to $7.21]$. The effect size was found to be negligible ($d < 0.01$), with a mean difference of only 0.04 points. Although visual inspection of the data suggested potential differences in variability between groups, a Levene's test for the equality of variances confirmed that these differences were not statistically significant, $F(1, 52) = 1.58$, $p = 0.194$.

These converging results from multiple statistical approaches provide strong evidence against H1. Contrary to the initial prediction, significantly higher presence scores were not observed for the hand-based interaction compared to the controller-based interaction.
**Table 2 Descriptive statistics for presence scores. The theoretical range for presence scores is 29 to 145.**

| Group | N | Mean | SD | Range |
|---|---|---|---|---|
| Hand-based | 27 | 108.56 | 14.69 | 81–136 |
| Controller-based | 27 | 108.59 | 11.65 | 90–131 |
| Overall | 54 | 108.57 | 13.13 | 81–136 |

**Table 3 Statistical comparison of presence scores between interaction groups.**

| Test | Statistic | df | p-value | 95% CI | Mean difference |
|---|---|---|---|---|---|
| Independent t-test | t = −0.01 | 49.4 | 0.992 | [−7.29 to 7.21] | 0.04 |

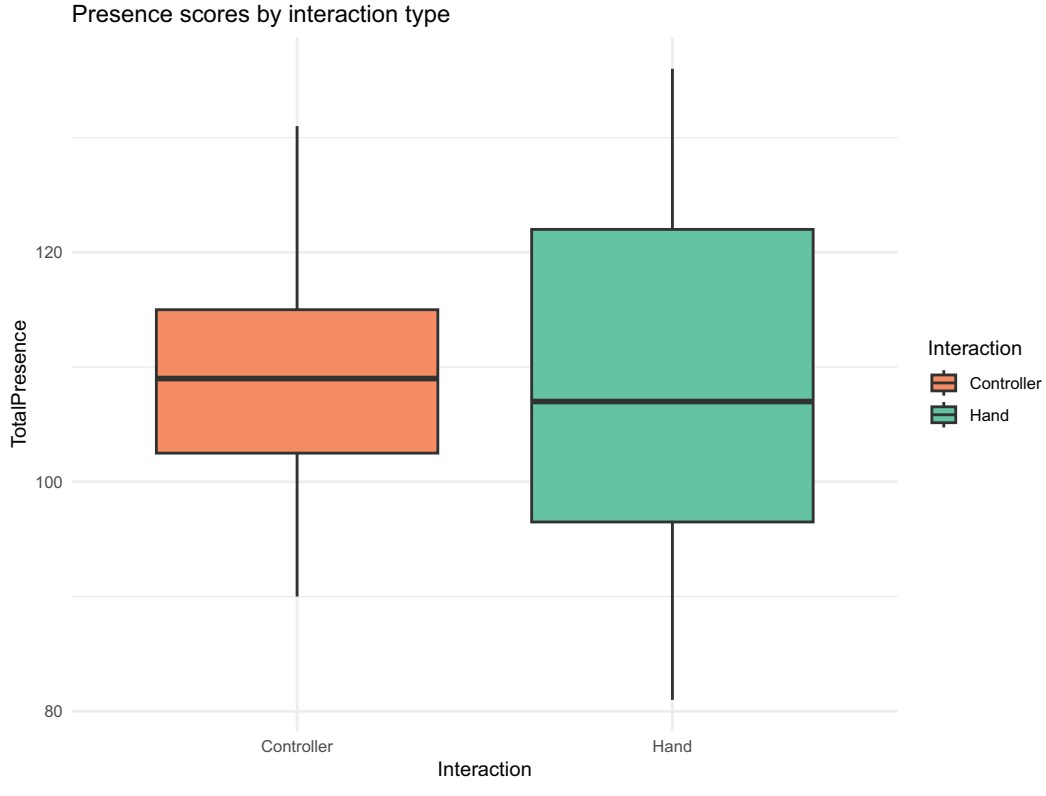

**Figure 8 Boxplot of presence scores by interaction group.**

Instead, the data suggest functional equivalence between the two interaction methods in terms of their ability to generate user presence. Detailed statistical results are presented in Table 3, and thus H1 is not supported.

**Visualization:** Fig. 8 presents a boxplot of presence scores by interaction group, illustrating the overlapping distributions and the lack of notable differences.

**Table 4 Descriptive statistics for performance scores.**

| Group | N | Mean | SD | Range |
|---|---|---|---|---|
| Hand-based | 27 | 20.26 | 2.97 | 14–24 |
| Controller-based | 27 | 21.30 | 2.89 | 16–24 |
| Overall | 54 | 20.78 | 2.95 | 14–24 |

**Table 5 Statistical comparison of performance scores between interaction groups.**

| Test | Statistic | df | p-value | 95% CI | Mean difference |
|---|---|---|---|---|---|
| Independent t-test | t = −1.30 | 52 | 0.199 | [−2.64 to 0.56] | 1.04 |

## Structured observation form analysis

This subsection addresses RQ2: how do hand-based and controller-based interactions affect interaction performance in educational VR simulations?

**Reliability:** the structured observation form demonstrated good internal consistency, with a Cronbach's alpha of 0.818, surpassing the reliability threshold.

**Descriptive statistics:** Table 4 provides the descriptive statistics for performance scores. The hand-based group had a mean performance score of 20.26 ($SD = 2.97$), slightly lower than the controller-based group's mean of 21.30 ($SD = 2.89$). The total sample mean was 20.78 ($SD = 2.95$).

**Group comparison:** an independent sample $t$-test revealed no significant differences in performance scores between the hand-based and controller-based groups, $t(52) = −1.30$, $p = 0.199$, 95% CI [−2.64 to 0.56], with a mean difference of 1.04. Detailed statistical results are provided in Table 5. In contrast to H2, controller-based interaction did not result in significantly better performance than hand-based interaction.

**Visualization:** Fig. 9 shows a boxplot of performance scores by interaction group, highlighting the similarity in the distributions despite a slight trend favoring the controller-based group.

## Supplementary analyses

**Effects of gender on presence scores:** an independent samples $t$-test revealed no significant effect of gender on presence scores, $t(51.9) = −0.70$, $p = 0.487$, indicating that gender did not influence participants' sense of presence.

**Effects of prior VR experience on presence scores:** prior VR experience, assessed through participants' self-reported familiarity with virtual reality technology, did not significantly affect presence scores, $t(37.4) = 1.159$, $p = 0.254$. This suggests that familiarity with VR technology was not a confounding factor in this context.

**Association between gender and previous VR experience:** a chi-square test examined whether gender and previous VR experience were associated. The analysis did not reveal a

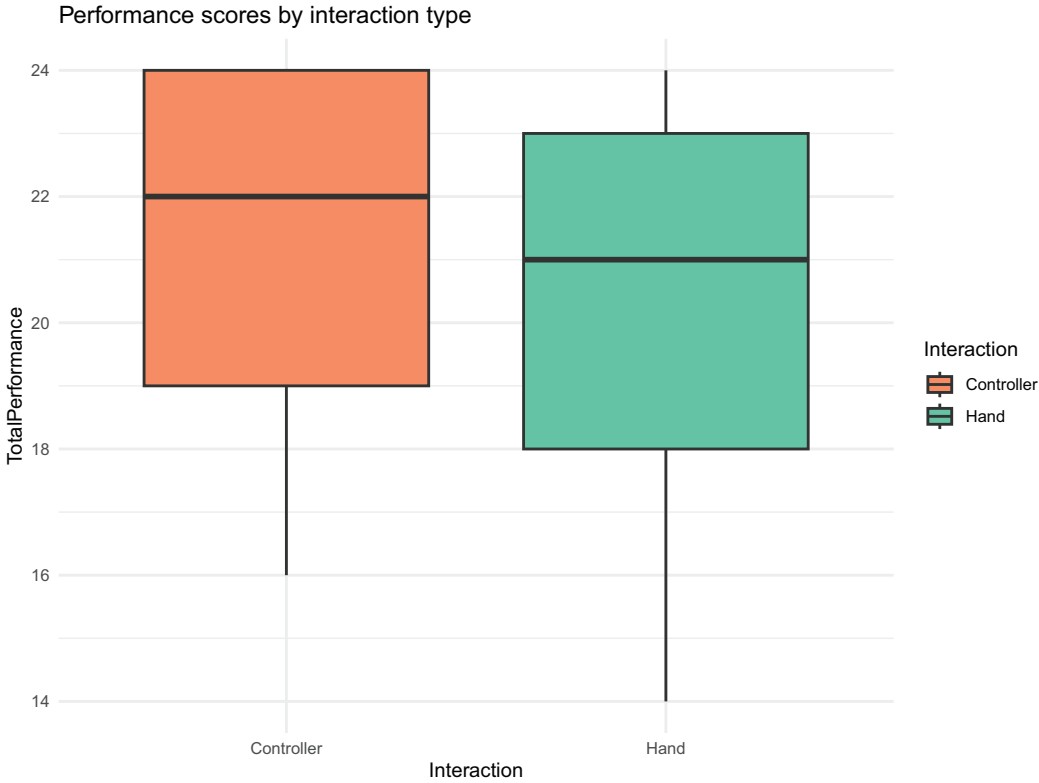

**Figure 9 Boxplot of performance scores by interaction group.**

significant association between gender and previous VR experience, $\chi^2(1) = 0.799$, $p = 0.371$, indicating that males and females had similar levels of previous VR experience in our sample.

**Effects of prior VR experience on performance:** an independent sample test $t$ did not show a significant effect of previous VR experience on performance scores $t(35.12) = 0.120, p = 0.905$, suggesting that prior familiary with VR technology did not influence task performance.

**Interaction effects (interaction type $\times$ gender):** A two-way ANOVA of the presence scores did not show a significant interaction between the type of interaction and the gender, $F(1, 50) = 0.020, p = 0.889$, nor a main effect of the gender, $F(1, 50) = 0.458$, $p = 0.502$. For performance scores, the interaction was also not significant $F(1, 50) = 0.367, p = 0.547$. However, a significant main effect of gender on performance scores was observed, $F(1, 50) = 4.844, p = 0.032$, suggesting that gender influenced performance independently of the type of interaction. An independent sample t-test was conducted to compare performance scores between men and women. The results indicated a significant difference in performance scores between males and females ($t(47.3)$ $= -2.222, p = 0.031$), with males having significantly higher performance scores. These results are detailed in Table 6.

**Table 6 Two-Way ANOVA results for interaction effects on presence and performance scores.**

| Measure | Effect | F(df) | p-value | Significance |
|---|---|---|---|---|
| Presence scores | Interaction (type × Gender) | $F(1, 50) = 0.020$ | 0.889 | Not significant |
| | Main effect (Gender) | $F(1, 50) = 0.458$ | 0.502 | Not significant |
| Performance scores | Interaction (type × Gender) | $F(1, 50) = 0.367$ | 0.547 | Not Significant |
| | Main effect (Gender) | $F(1, 50) = 4.844$ | 0.032 | Significant |

**Controlling for confounding variables:** to address possible confounder effects, an ANCOVA was performed with performance as a dependent variable, gender as the main factor, and previous VR experience as a covariate. The analysis confirmed that the effect of gender on performance remained significant even after controlling for previous VR experience, $F(1, 51) = 4.191$, $p = 0.046$, while previous VR experience did not show significant effect, $F(1, 51) = 0.049$, $p = 0.826$. This indicates that the observed gender difference in performance cannot be attributed to differences in prior VR experience.

## DISCUSSION

This study investigated the impact of hand-based *vs.* controller-based interaction methods on sixth-grade students' sense of presence and interaction performance within a VR science learning environment. Contrary to initial hypotheses, the results did not reveal significant differences in presence or interaction performance between the two interaction groups. Although hand-based interaction was hypothesized to enhance presence (H1) and controller-based interaction was expected to improve performance (H2), neither of these predictions was statistically supported by the data. These findings will be discussed in light of existing literature and theoretical frameworks, particularly considering the sensory contingency theory of presence and cognitive load theory, alongside the supplementary findings of this study.

### Presence in hand-based and controller-based VR interaction

Hypothesis 1, which posited that hand-based interaction would lead to a significantly higher sense of presence compared to controller-based interaction, was not supported. Statistical analysis did not reveal significant differences in presence scores between the two groups, with both groups reporting comparable mean presence levels. This finding contrasts with the theoretical expectation that the more direct and natural sensorimotor contingencies provided by hand-based interaction would enhance the subjective feeling of 'being there' (*Slater, 2018*; *Flanagan et al., 2001*).

Several factors may explain the finding of equivalent effectiveness between interaction methods. Firstly, as *Witmer & Singer (1998)* highlighted, *sensory fidelity* and *control* are crucial determinants of presence. It is possible that the VR system employed in this study, regardless of the interaction method, provided a sufficient level of sensory fidelity and responsiveness to elicit a comparable sense of presence in both groups. This suggests that for educational VR applications targeting sixth-grade students, both interaction modalities can achieve the threshold necessary for effective immersive learning experiences.

Furthermore, while hand-based interaction is often lauded for its naturalness and intuitiveness (*Lin et al., 2019*; *Lou et al., 2021*), the current limitations of hand-tracking technology, as noted in the literature review, including potential *unstable tracking* and *decreased precision* (*Argelaguet et al., 2016*; *Kilteni, Groten & Slater, 2012*; *Tong et al., 2023*), may have balanced out the anticipated advantages in presence. Participants in the hand-based interaction group might have experienced moments of frustration or reduced agency due to these technological limitations, thereby offsetting any potential increase in presence derived from the more direct interaction style. This technological equilibrium between the intuitive appeal of hand-based interactions and the reliable precision of controller-based interactions may explain the comparable results observed in this study.

In contrast, controller-based interaction, while less directly embodied, offers a reliable and familiar mode of control within VR environments. Participants, especially those with previous gaming experience, might find controllers intuitive in their own right, providing a sense of *control* that is equally conducive to presence (*Khundam et al., 2021*). Moreover, as *Fitton, Finnegan & Proulx (2020)* discovered, mere immersion in a virtual environment, even when replicating a traditional classroom setting with PowerPoint content, could achieve levels of engagement and presence comparable to physical classrooms, suggesting that the virtual context itself is a potent factor in eliciting presence, possibly overshadowing the nuances of interaction modality in this study.

It is also important to consider the nature of the presence measure used in this study. Self-report questionnaires, while widely adopted (*Witmer & Singer, 1998*), may not fully capture the subtle differences in the qualitative experience of presence that different interaction methods might offer. Future research might benefit from incorporating more nuanced measures, including physiological responses or qualitative interviews, to further explore the subjective nuances of presence related to interaction modalities.

## Interaction performance with hand-based and controller-based VR interaction

Hypothesis 2, which predicted that the controller-based interaction would result in significantly better interaction performance compared to the hand-based interaction, was also not supported. Structured observation form analysis did not reveal significant differences in performance scores between the two groups. This finding is somewhat unexpected given that controller-based interactions are often associated with higher precision and efficiency in VR tasks, especially those that require object manipulation and specific actions (*Caggianese, Gallo & Neroni, 2019*; *Khundam et al., 2021*).

According to Cognitive Load Theory (*Mayer, 2001*; *Sweller, 2011*), instructional design should aim to minimize the extraneous cognitive load to optimize learning. We hypothesized that controller-based interaction, which is potentially more familiar and offers more direct input for certain types of interaction within digital interfaces, might reduce extraneous cognitive load related to task execution, leading to improved performance. However, the results suggest that the cognitive load imposed by either interaction method in the context of these science experiments did not differ significantly in terms of affecting overall performance.

One possible explanation is that the design of the virtual science experiments in this study was not sufficiently complex or precision-demanding to highlight the potential performance advantages of controllers. Although controllers are often favored for tasks that require high precision or ray casting (*Lou et al., 2021*; *Luong et al., 2023*), science experiments might have focused more on conceptual interaction and exploration than tasks that require fine motor skills or rapid and precise manipulations. In simpler interactive scenarios, the intuitiveness of hand-based interaction, despite potential tracking limitations, might have compensated for any inherent precision advantage of the controllers. Furthermore, while the structured observation form was designed to capture key performance aspects, it might not have been sensitive enough to detect subtle differences in task efficiency or error rates that could have favored one interaction method over the other.

It is also possible that the learning curve associated with hand-based interaction, if any, was quickly overcome by sixth-grade students in the context of these engaging and intrinsically motivating science experiments. The novelty and embodied nature of hand-based interaction might have fostered a different type of engagement that, while not necessarily improving presence scores in our questionnaire, allowed participants to effectively interact with virtual experiments and achieve performance levels comparable to their controller-using counterparts. As suggested by *Johnson-Glenberg (2018)*, gesture-based VR systems can boost engagement and educational results by directly engaging sensorimotor pathways, which could have played a role in performance in the hand-based group, even without a measurable presence difference.

## Supplementary findings and considerations of the effect of gender and previous experience

Supplementary analyses did not reveal significant effects of previous VR experience on presence scores, reinforcing the primary findings on interaction modality. However, a closer examination of the performance scores revealed a notable gender difference, with males demonstrating significantly higher performance than females.

Addressing the potential confounding role of prior VR experience—as this could theoretically explain performance differences if one gender had systematically more VR exposure—our analysis revealed no significant association between gender and prior VR experience in our sample. Furthermore, the previous VR experience itself did not show a significant effect on performance. Most importantly, when controlling for previous VR experience through ANCOVA, the effect of gender on performance remained statistically significant, confirming that the observed performance gap cannot be attributed to differential familiarity with VR between genders.

This result, since the literature review did not show a clear indication of this finding, warrants further investigation. The persistence of the gender effect after controlling for prior VR experience suggests that other factors may be at play, such as differences in spatial cognition, motor control strategies, technology anxiety, or task-specific skills that may manifest differently in virtual environments. More research is needed to explore the

underlying mechanisms for this gender disparity and to determine its generalizability across other VR learning environments and interaction methods.

## Limitations of the study

The focus of the study on sixth graders and specific tasks (for example, science experiments) limits the generalizability of the findings. These tasks, while educationally relevant, may not reflect scenarios requiring greater precision, such as technical training. Furthermore, the short duration of the study provides limited information on long-term adaptation, a critical factor for sustained VR use (*Akyuz et al., 2020*).

A further limitation concerns the adaptation of the Turkish Presence Questionnaire. Although the core structure and content of all 29 items were retained, minor wording adjustments were made to a few interaction-related questions to reflect the specific input method experienced by each group (hand-based or controller-based interaction). These changes were implemented to support more accurate self-assessments and improve linguistic clarity in Turkish. However, such modifications may have introduced a subtle bias by drawing participants' attention to the type of interaction, potentially influencing their responses. Although the adapted versions showed high internal consistency (Cronbach's $\alpha = 0.832$ for both versions, compared to the original's $\alpha = 0.844$), the possibility of response bias due to question framing must be acknowledged. Future research should consider alternative ways to ensure clarity without explicitly emphasizing group conditions in questionnaire items.

Finally, reliance on structured observations introduces methodological limitations. Although the observation form showed acceptable reliability (Cronbach's $\alpha = 0.818$), using a teacher familiar with the students to perform assessments raises concerns about potential bias. The subjective nature of these ratings, combined with challenges in observing interaction performance through a live feed (often providing limited or blocked views), may affect the objectivity and replicability of the data. Future studies could incorporate validated tools like the System Usability Scale (SUS) alongside objective measures, such as task completion times, to reduce bias and enhance performance analysis.

## CONCLUSION

This study explored the impact of hand-based *vs.* controller-based interactions in a VR educational environment designed for sixth-grade students. The results did not reveal significant differences in presence or performance between the two interaction methods, challenging the common assumption that one approach inherently outperforms the other. For sixth graders, both methods appear equally effective, providing educators and VR developers with valuable flexibility. This finding shifts the focus from presumed superiority to practical considerations such as cost, maintenance, and hygiene when designing VR systems for educational settings.

One notable advantage of hand-based interactions is their ability to address logistical challenges tied to shared controllers. Controller-based systems require regular maintenance, such as battery replacements, and pose hygiene concerns with multiple users handling the same devices. In contrast, hand-based interactions leverage natural hand

movements, eliminating these issues and reducing maintenance efforts. However, implementing hand-tracking technology requires more sophisticated software and hardware capable of reliable hand tracking, which may involve higher initial costs or technical complexity compared to controller-based systems. For example, devices such as the Oculus Quest 3, used in this study, include built-in hand-tracking capabilities, mitigating some of these concerns by eliminating the need for additional hardware purchases. Although this makes hand-based interactions appealing for resource-constrained schools aiming to integrate VR into their curricula, schools must consider the trade-offs between initial setup complexity and long-term maintenance savings. Crucially, the results show that these logistical benefits do not come at the cost of performance; the study did not support H2, as controller-based interaction failed to yield significantly better interaction performance than hand-based interaction.

Beyond practical benefits, the study revealed a notable influence of gender on performance, independent of interaction type. This suggests that individual differences, possibly related to spatial skills, technological familiarity, or motor abilities, play a critical role in VR effectiveness. Such findings highlight the need for an inclusive VR design. To ensure equitable learning opportunities, developers should prioritize adaptable interfaces or diverse interaction options that accommodate varying user profiles. By addressing these differences, VR systems can prevent disparities, particularly gender-based ones, and create a more accessible educational tool for all students.

To build on these findings, future research should expand in several key directions. First, testing hand-based and controller-based interactions across diverse age groups and educational contexts will enhance the generalizability of the results, providing a clearer understanding of how different demographics engage with VR. For example, younger learners might adapt more quickly to hand-based controls, while older students could prefer controllers due to familiarity. Second, incorporating objective metrics, such as task completion times or error rates, alongside observational data will offer a more robust and unbiased assessment of performance, reducing the reliance on subjective interpretations and strengthening the analysis. Third, conducting longitudinal studies is essential to evaluate how students adapt to VR interactions over time, revealing insights into sustained engagement, learning retention, and skill development that short-term studies might miss. Finally, investigating how individual differences, such as gender, previous technology experience, or cognitive abilities, shape VR outcomes will be critical to designing tailored systems that meet diverse user needs, ensuring VR tools are inclusive and effective across diverse populations. Together, these research avenues address current limitations and lay the groundwork for more impactful VR educational tools.

In conclusion, this research advances the field of educational VR by demonstrating that hand- and controller-based interactions effectively support presence and performance among sixth-graders. It underscores the value of practical considerations—cost efficiency and hygiene—and the need for inclusive design to address user diversity. Although limitations in scope and methodology temper broader claims, they also pave the way for future exploration. As VR continues to transform education, refining interaction methods

and understanding their impact across diverse student populations will be critical to realizing its full potential as an innovative and equitable learning tool.

## ACKNOWLEDGEMENTS

The author would like to acknowledge the use of **Writefull** for grammar and spell checking. The tool was integrated into **Overleaf** and employed throughout the writing process to enhance the clarity and correctness of the manuscript.

### Funding
The authors received no funding for this work.

### Competing Interests
The authors declare that they have no competing interests.

### Author Contributions
- Murat Saran conceived and designed the experiments, performed the experiments, analyzed the data, performed the computation work, prepared figures and/or tables, authored or reviewed drafts of the article, and approved the final draft.

### Ethics
The following information was supplied relating to ethical approvals (*i.e.*, approving body and any reference numbers):

Çankaya University Science and Engineering Sciences Scientific Research and Publication Ethics Committee granted approval to carry out the study (Ethical Application Ref: 31.05.2025/9)

### Data Availability
The data and code are available at Zenodo: msaran1923. (2025). msaran1923/VR_pub_data: Code and Data (publish). Zenodo. https://doi.org/10.5281/zenodo.14992040.

### Supplemental Information
Supplemental information for this article can be found online at http://dx.doi.org/10.7717/peerj-cs.3168#supplemental-information.

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
