# Peer review of "Comparing hand-based and controller-based interactions in virtual reality learning: effects on presence and interaction performance"

_PeerJ Computer Science, doi:10.7717/peerj-cs.3168_

## Round 0.1 · original submission · Major Revisions

Thank you for sending your contribution to PeerJ in Computer Science on an up-to-date topic such as virtual reality for educational applications.

The paper is clear and well written. However, there are some major concerns that need to be addressed.

Please, read carefully the Reviewers’ comments and provide an answer to all their concerns with a point-by-point response. Especially consider

1- Revising the introduction according to the provided comments.
2- Providing a correct structuring of research questions and hypotheses.
3- Using proper scientific source of information and more recent papers on the topic to ameliorate the state-of-the-art.
4- Revising some statements that may need to be mildened according to the Reviewers’ comments.
5- Addressing the concerns related to the experimental design and the possible presence of bias.
6- Providing an extended discussion on the results and of the study limitations, checking also on the possible real-world costs of employment of this new system.

Reviewer 1 ·

Basic reporting

The article presents an experimental study comparing hand-based and controller-based interaction in virtual reality, focusing on perceived presence and task performance in a well-balanced sample of 54 sixth-grade students. The manuscript is generally well structured and clearly written, with a relevant and timely research topic. However, there are several major aspects that would benefit from further improvement.

First, the acronym "VR" should be introduced once—preferably in the opening paragraph—and used consistently thereafter. In the current version, there are several instances where both the acronym and the full term "virtual reality" are used interchangeably or redundantly, which can disrupt the flow and readability.

Then, the introduction could be made more concise and focused. While it covers a wide range of concepts - including a general definition of VR, market growth, and applications in education - some of these sections could be streamlined or tied more directly to the specific objectives of the study. For example, the section discussing the growth of the VR market (lines 56-75) includes references to several specific devices that may quickly become outdated. A more impactful and lasting way to illustrate the growth of the field might be through data visualizations, such as the trend in VR-related academic publications over the past decade.

In addition, the inclusion of sources such as Wikipedia is generally discouraged in academic writing unless absolutely necessary, and even then, such references are best placed in footnotes rather than cited in the text.

The transition to discussing the use of VR in education (line 76) is generally well-placed. However, it is followed by a return to technical aspects—such as high resolution and low latency—which somewhat disrupts the logical flow of the section. To improve clarity and coherence, I would suggest reducing the emphasis on technical advancements and possibly confining them into a dedicated, standalone paragraph earlier in the introduction.
Moreover, at line 76 you mention "virtual reality also offers significant potential in education". This sentence would benefit from additional supporting references. It's also worth noting that the effectiveness of VR for educational outcomes is still under debate, as you reference at line 122 "VR [...] led to lower transfer test performance due to elevated extraneous cognitive load and decreased cognitive engagement" and line 168 "These findings indicate that VR’s influence on student learning is complex". Therefore, claims such as "thus deepening their engagement and improving learning outcomes" (lines 78-79) could be presented with a more cautious tone.

The subsection on the impact of virtual reality on student learning is appropriate, but not completely relevant to the study and could be reduced, which is about interaction modalities in virtual reality, rather then learning (which is not assessment by any questionnaire and is not reported in the results).

First, when designing an educational activity, it is essential to clearly define the learning objectives and ground the design in a well-established theoretical framework. While the section refers to the Cognitive Theory of Multimedia Learning, it remains unclear whether this is the theoretical lens the authors have fully adopted, or whether other complementary or alternative models-such as the ICAP framework, Bloom's Taxonomy, or constructivist theories-also informed the development of the learning task. Clarifying this point would strengthen the theoretical foundation of the study.
Similarly, you mention Cognitive Load Theory at line 609, even though it is not mentioned anywhere previously.

In addition, the reference to embodied agents in this subsection seems somewhat out of place. It may be more appropriate to discuss this topic in the "Construct of Presence" section, where it would be more in line with concepts such as embodiment and interaction within the virtual environment.

On that note, I particularly appreciated the explanation of presence through the lens of Witmer and Singer's work. However, incorporating Slater's perspective-particularly his work on embodiment (e.g., The Sense of Embodiment in Virtual Reality, Slater, 2012)-could provide a richer and more detailed discussion, especially given the relevance of embodiment to the focus of the study on interaction modalities.

Finally, while the section presents two different definitions of immersion, it is not entirely clear which of these definitions the authors adopt for their study. A clearer position on this point would help to align the theoretical framework with the research design, and improve the overall coherence of the manuscript.

A concluding remark pertaining to the introduction is related to the statement of research hypothesis.
At line 296 you mention "Research supports this hypothesis by highlighting the immersive and natural qualities of hand-based interactions". While this is a relevant and promising claim, it would benefit from one or more supporting references to strengthen its foundation.

Experimental design

Methodologically, the study is well organized and clearly presented. The research questions and hypotheses are explicitly defined. However, a shorter and more clear formulation of the hypotheses could help improve clarity and focus.

The subsection on participants is well articulated, with a balanced and appropriately sized sample.

I was particularly impressed by the careful description of the learning activities. The integration of images and the effort to replicate the real laboratory environment in the virtual space show great attention to detail and thoughtful design aimed at increasing the sense of presence. In this regard, including an image of the real lab alongside the virtual replica (e.g., in Figure 3) could have further enhanced the impact of your description and visually underscored the fidelity of the virtual environment. In addition, for Figure 1, it would be useful to clarify whether it is an original illustration or whether credit should be given to an external source.

Regarding the administration of questionnaires and data collection, I have some concerns. As a general recommendation, it is recommended to avoid modifying or tailoring items from validated questionnaires unless absolutely necessary (see line 436). Modifications can affect reliability and comparability with previous research.
More critically, the use of an observation form scored by a teacher familiar with the students raises concerns about potential bias. The subjective nature of such ratings, combined with the challenge of observing interaction performance through a live feed (which often has limited or blocked views), can affect the objectivity and replicability of the data. This could affect the validity of the results. The use of validated tools - such as the System Usability Scale (SUS) - could help to mitigate this issue. This limitation should be explicitly acknowledged in the appropriate section.

Validity of the findings

The concerns related to repeatability and potential bias in the results are appropriately acknowledged in the Experimental Design section.
As for the results, they are clearly reported and well-structured. However, I would like to raise one major point for consideration. The finding that no significant differences emerged in terms of presence and interaction performance across the two conditions is quite meaningful, and perhaps deserves to be framed as such, rather than being referred to as a “null finding” (line 577). Even in the absence of statistically significant differences, these outcomes can still carry important implications, especially when considering the broader context of user interaction design in educational VR.
You also report a significant difference in performance based on gender, which is indeed an interesting result. That said, I would caution against overinterpreting this without further contextual analysis, as correlation does not imply causation. In this context, one potentially confounding factor that might explain the observed performance gap could be prior VR experience—yet I did not find an analysis exploring the relationship between participants' previous exposure to VR and their interaction performance.

Additional comments

No comments

Reviewer 2 ·

Basic reporting

The paper presents an empirical study involving sixth-grade students to test their sense of presence and interaction performance in a VR science laboratory simulation.
Three interactive science experiments have been performed by 54 students, divided into 2 groups (balanced in number of male and female), so that one group performs the tasks with direct interaction, and the other one exploiting the controllers.
Statistical analysis has been performed to check if there is statistical difference in the presence and in the performances achieved, and a second analysis regarding differences between male and female.
The paper is well written and easy to follow, and well covers the state of the art. The results, even if does not support the initial hypotheses, are interesting and useful for future research. Some parts need some improvements (see general comments), but in the overall the paper is well organized and results well described.

Experimental design

The proposed experiment is a between study (I suggest to clearly state it in the setup description) to see if the type of interaction (direct vs controller) used by the participants affects the results in terms of presence and performances.
I report later on some comments regarding the data analysis, that misses some information.
A study of the time needed to perform the tasks may improve the analysis of the performances, and similarly a deeper comparison regarding some key questions from both the questionnaires may improve the insight of your results. Authors have the ok of the university ethical committee

Validity of the findings

I suggest to add a specific “threat to validity” section where to discuss the limitations of your experiment (not to do this in the conclusion). As already identified by the authors, these results cannot generalize too much, above all to students of different age, but can guide in further experiments and well integrate in the state of the art

Additional comments

I report later on some specific comments:
• “Wikipedia contributors, 2025”: il possible, please avoid citing wikipedia, and refer to academic publications or websites of the producers of these devices
• Section Methods: usually the organization of such section is the contrary, you start from the RQs, and then you made hypotheses regarding the answers. Please reorder this part
• “The students were randomly divided into two groups”: did you try to balance male and female? otherwise an unbalanced distribution may end in a new factor... it seems that there is balance reading later on, ok, but please clarify this aspect here too. It is ok selecting randomly, but you should ensure gender balance in this case
• “specific questions related to interaction and control were tailored to reflect the input method experienced by participants.”: I do not understand why you changed the questions in such a way. The generic one was coherent with both the groups, while modifying them in this way may add a bias, since it is clear from the question that you are focusing on the type of interaction, pushing the subject to focus on this, which should be avoided because can create bias. Maybe you should discuss it in the threat to validity session
• “Participants indicated their gender by selecting either ”Male” or ”Female” using a checkbox format”: forcing a subject to declare the gender is usually not appropriate... you should include a third option like "prefer not to tell" or something similar, even if it may cause you to exclude that subject from some result analysis (for example the one regarding gender)
• “The questionnaire was administered immediately after each participant completed the VR learning task to capture their immediate impressions, enhancing the precision of their self-reported experiences””: it was the presence questionnaire to be filled after each task, correct? In VR or not? If in VR ,maybe a screenshot can be interesting. Also, please include in the additional material the questionnaire filled by the teacher too.
• “Participants completed a paper version of the questionnaire”: it is not clear, again the same questionnaire? or which one, the demographic one? Please explain better which questionnaires are filled and when
• Table2: could you please report the min and max value that should be achieved? so that to understand if the mean is a good, or not, result
• “In contrast to H1, hand-based interaction did not produce higher presence scores than controller-based interaction”: I'm not too expert in statistical analysis, but as far as I remember, there should be an analysis comparing not only the means, but the overall distribution of two sets: the means are similar, it is true, but from the box plot seems to be an higher distribution in one group wrt the other. So, a comparison between the 2 populations should give you more information, and if there is some other statistically significant differences. Or if this analysis is already done as requested, and is the one you are describing here, please explain this better because it seems to refer instead to the means only
• Maybe I lost this detail somewhere, but I cannot find the type of device you used (a meta quest, a Vive…?): if not present, could you please add this information? Also, seems to be that you did not report how long the task requires, more or less, and the total duration of the experiment, please report this information
• “Supplementary findings and considerations of the effect of gender and previous experience”: please also consider that there may be a difference in the preparation / skills between males and females. Unfortunately, there is still a big gap in how STEM disciplines are proposed to male and female (in schools but in general in the society), so for example if a similar experiment is conducted not with VR, but in a real scenario, such discrepancy may anyway arise. Such kind of test should be for example also a verification of this result: if you ask these students to perform similar tasks in their lab, is this gender difference in results still present...? so that to understand if it is related to VR or not. Also, as suggested in the conclusion, if you think that there could be possibly differences in spatial skills, maybe you should integrate the discussion section with some literature regarding this, so that to better analyze this difference
• “This makes them particularly appealing for resource-constrained schools that want to integrate virtual reality into their curricula”: here you should consider the complexity of exploiting hand tracking too, because it requires a more complex software, and also a hardware able to perform hand tracking, that usually is more expensive…

---

## Round 0.2 · Minor Revisions

I thank the Author for having provided a clear point-by-point response to the Reviewers’ concerns as well as mine.

The manuscript has greatly improved and there are just a couple of minor issues to address.

Besides providing a new point-by-point response to the Reviewers’ comments, particularly consider addressing the following major concerns:

- Check the source of Figure 2 and ensure to properly cite it in the text. If yours is a modified version of a pre-existent image, please specify it and refer to the original image.
- Highlight the concept of presence in VR, where requested by Reviewer 2.
- It seems that some of the explanations given in the response are satisfactory for the Reviewers but are not completely reported in the new manuscript version.
- Please, consider clarifying some statements allowing a better readability and interpretability of your work and results.
- Check the manuscript for typos and formatting issues.

Reviewer 1 ·

Basic reporting

The author has successfully addressed all the raised concerns. Only one minor issue remains: an acronym in line 56 that was not corrected.
Additionally, there is a small concern regarding Figure 2, which appears to be modified from an external source and may require proper attribution or clarification. Could you please provide a clarification about it? https://www.roadtovr.com/magic-leap-2-controller-camera-inside-out-tracking/

Experimental design

No comment.

Validity of the findings

No comment.

Additional comments

No comment.

Reviewer 2 ·

Basic reporting

I thanks the author for the accurate review and answer, which faced quite all of my comments, where possible. The paper has improved, I report later on only some minor points to be fixed

Experimental design

Improved as requested

Validity of the findings

Improved as requested

Additional comments

I report later on some specific comments/typos:
- I read the version with removed/addtions, and I see some problems in formatting (lines going “longer” than the column, “.” Far from text etc), but this could be related to this version of the paper: anyway, please check in the final version that there problems are no longer there
- Page2, lines 90-93: I do not know why you removed this part (probably to answer to the other reviewer, and this I s fine), but something on the presence in VR I think is important here, considering the topic of the paper (here or later on, please check to still have something on this)
- what is K-6 education...? please provide an explanation
- “participants were randomly assigned to one of two groups: One group used traditional control devices while”…: please, after : do not use upper case, and above all do not stop the sentence with a ".". Both groups should be described after the :
- usually, in the introduction you report briefly the results you obtained, and present a summary of the sections of the paper (why aren't the sections numbered...? if it is not forbidden by the journal format, I strongly suggest to enumerate each section, so that you can simply refer to them in the text)
- “the following hypotheses were formulated”: I would add “to answer the RQs”
- Line 454, please add here too the fact that you balanced gender
- In “Results” (why is this all upper case?), please specify where you ar answering RQ1 and where RQ2
- Please clearly state that H2 is not supported by the results (I would add thid at the end of line 764)
- Lines 771-772: please check this sentence, it is not clear: "similar to previous VR experience" but you did not discussed previous VR experience yet
- Line 772 “presence scores did not significantly affect”: were not significantly affected (? Please check this sentence)

---

## Round 0.3 · accepted · Accept

Thank you for sending the revised version of the manuscript and for addressing all the Reviewers’ concerns.

Please, notice that Reviewer 1 has not been invited in this round of revision, having asked for very minor changes. I have assessed Reviewer 1’s required changes and verified that they were correctly applied.

Reviewer 2 is now satisfied with the current version of the manuscript.

I just point out to the Editorial Staff that there is a remaining typo at line 575: “(3) additional analyses” should be “additional analyses”.

The manuscript is now ready for publication.
Thank you for your work and for sending your study to PeerJ in Computer Science.

Reviewer 2 ·

Basic reporting

thanks for the update to the paper, that now seems to me ready for publication

Experimental design

ok

Validity of the findings

ok